# COMMON 7B LANGUAGE MODELS ALREADY POSSESS STRONG MATH CAPABILITIES

## ABSTRACT

It was once believed that mathematical capabilities in language models required either large model scales or extensive math-related data pre-training. However, this paper demonstrates that the small-scale LLaMA-2 7B model already possesses strong mathematical potential. This is evidenced by its impressive scores of 97.6% on GSM8K benchmark and 70% on MATH benchmark, achieved by selecting the oracle response from 1024 generations. Equipped GPT-4 Turbo as an additional verification, LLaMA-2 7B also achieves 91.8% accuracy on GSM8K benchmark. This indicates that the primary issue within current models is the difficulty in consistently eliciting the inherent mathematical capabilities. We find that scaling up synthetic SFT data, which proves to be nearly as effective as real data, can significantly enhance the reliability of generating correct answers. Surprisingly, even with approximately one million samples, we observe no clear performance saturation. And our method is more efficient with large data scale than previous works. This approach achieves an accuracy of 82.4% on GSM8K and 40.1% on MATH using LLaMA-2 7B model, surpassing GPT-3.5 Turbo. Our 70B model even exceeds an early version of GPT-4 on MATH and out-of-domain Hungarian National High School Math Exam. These results demonstrate our method significantly elicits the general mathematical capabilities of language models. Also, we provide insights into scaling behaviors across different reasoning complexities.

## 1 INTRODUCTION

Mathematical capabilities have traditionally been viewed as so challenging that they are believed to emerge in common language models only when these models reach a very large scale. For instance, studies by Wei et al. (2022a;b) suggest that only models with size exceeding 50 billion parameters can attain decent accuracy. This statement is proved by the observation that when testing small models on mathematical reasoning benchmarks such as GSM8K Cobbe et al. (2021) and MATH Hendrycks et al. (2021), the accuracy is significantly low, with LLaMA-2 7B model achieving only 14.6% and 2.5% respectively Luo et al. (2023). A natural strategy to equip smaller language models with mathematical abilities involves creating math-specific base models trained on hundreds of billions of math-related data Lewkowycz et al. (2022); Azerbayev et al. (2023); Shao et al. (2024) since it directly enrich the model's repository of background knowledge. However, the accuracy of such models remains modest; for example, Llemma-7B Azerbayev et al. (2023) only achieves 36.4% on the GSM8K benchmark and 18.0% on the MATH benchmark.

In this paper, we present a counterintuitive observation that general pre-trained language models of small sizes, such as LLaMA-2 with 7B parameters Touvron et al. (2023b), already possess strong intrinsic mathematical capabilities. To substantiate this point, we first assess the limits of mathematical reasoning capabilities in pre-trained models. We employ the standard 1-shot setting to prompt the LLaMA-2 7B base model, generating 1024 answers for each question with different random seeds. Surprisingly, we find that 97.6% of GSM8K benchmark questions Cobbe et al. (2021) and 70.0% of MATH benchmark questions Hendrycks et al. (2021) can be solved within the 1024 generated responses. It is worth noting that these accuracies have even outperformed those of the GPT-4 model (92.0% on GSM8K and 42.5% on MATH) Achiam et al. (2023). The performance of GPT-4 models are continuously being improved. The GPT-4 Turbo (1106) API has increased accuracy to 94.8% on GSM8K and 64.5% on MATH. However, LLaMA-2 7B model using 1024 generations still outperforms it. We further incorporate GPT-4 Turbo to verify the correctness of

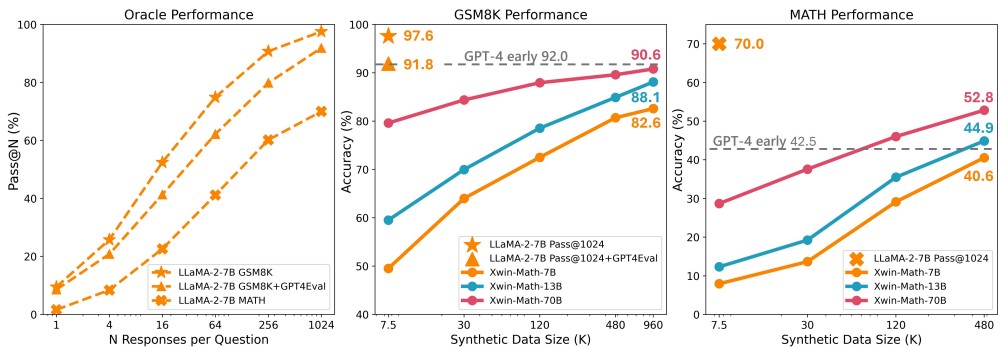

Figure 1: The orange star (97.6%, middle) and cross (70.0%, right) represent the accuracy achieved by selecting the oracle response from N=1024 generations of the LLaMA-2 7B model using ground truth. Incorporating GPT-4 Turbo for a more accurate assessment, the model still achieves a Pass@1024 of 91.8% on the GSM8K benchmark. Moreover, as the number of N increases, the Pass@N results continue to show significant improvement (left). The scaling experiments in the middle and on the right are done with separate synthetic datasets.

the intermediate reasoning steps, and the accuracy of GSM8K remains impressively high at 91.8%. Therefore, we conclude that LLaMA-2 7B model has indeed developed strong mathematical potential. The primary issue is the lack of guarantee that the correct answer will be dug out, as most generations are incorrect.

Table 1: Comparison of SFT data scaling with real versus synthetic math questions. All responses in training data are generated by GPT-4 Turbo for fair comparison. It reveals that synthetic math questions are nearly as effective as real ones.

| Data size | GSM8K-real | GSM8K-synthetic | MATH-real | MATH-synthetic |
|---|---|---|---|---|
| 0.94K | 26.7 | 25.9 | 4.2 | 3.9 |
| 1.88K | 32.8 | 31.9 | 5.6 | 4.9 |
| 3.75K | 43.3 | 42.2 | 6.6 | 6.0 |
| 7.50K | 50.2 | 49.5 | 8.4 | 7.9 |

To address this ranking issue, we observe that the accuracy improves almost in linear or even super-linear with exponentially increased supervised fine-tuning (SFT) data. Moreover, we note that the accuracy is far from reaching a plateau when utilizing all available GSM8K and MATH training data (as shown in Table 1). This observation encourages us to further scale up the SFT data. However, we face a challenge as there is a lack of real data to support this continuous scaling.

As a consequence, we turn to synthetic data, employing a high-performing language model, namely GPT-4 Turbo, to produce synthetic math questions. Although synthetic math data has been utilized in previous works Luo et al. (2023); Li et al. (2023); Yu et al. (2023), the challenge of generating large quantities of high-quality synthetic data remains unresolved. We find that a straightforward "brand-new" generation strategy, which prompts the GPT-4 Turbo to create a completely new question based on preference ones and then applies a simple verifier (also GPT-4 Turbo based), has been highly effective. The data collected using our approach exhibits significant gains over previous methods under the same conditions. Specifically, as indicated in Table 1, the use of synthetically generated math questions can achieve accuracy nearly on par with that of real questions, highlighting the potential of synthetic SFT math questions for the scaling purpose.

Leveraging synthetic data has allowed us to scale our SFT data significantly, for instance, from 7.5K to 960K on GSM8K and from 7.5K to 480K on MATH, respectively. This data scaling shows excellent scaling behavior, as drawn in Figure 1. Specifically, by simply scaling the SFT data, our model has become the first to exceed 80% and 40% accuracy on GSM8K and MATH, respectively, using a

standard LLaMA-2 7B base model (achieving 82.4% and 40.1% respectively)[1]. The straightforward synthetic SFT data proves effective from stronger base models as well, such as LLaMA-2 70B, which achieves 90.6% on GSM8K and 51.9% on MATH. To the best of our knowledge, this is the first open-source model to exceed 90% accuracy on GSM8K. It is also the first open-source model to approach GPT-4 (i.e., GPT-4-0314), demonstrating the efficacy of our simple synthetic scaling method. Our 70B model also achieves 75% (outperforming GPT-4 by 7%) on out-of-domain Hungarian National High School Math Exam, further showing that this method can significantly enhance the model's general mathematical capabilities.

In addition to the strong results, we have also gleaned insights into the effectiveness of our approach:

- As the scale of SFT data increases, the model's accuracy tends to plateau when utilizing 256 attempts; however, there is a marked increase using 1 response. This indicates that while the model's upper capability limit remains fairly constant, the performance gains are primarily due to enhanced stability in generating correct answers.

- The accuracy of solving math problems follows a power law with respect to the number of chain-of-thought (CoT) steps with different SFT data quantities. An expanded SFT dataset improves the reliability of each reasoning step. Further increasing the proportion of training samples with longer CoT steps through resampling can significantly improve the accuracy of difficult questions.

- Experiments on decontamination and out-of-domain benchmarks demonstrate that this method does not suffer from information leakage.

## 2 EXAMINE MATH CAPABILITY OF LANGUAGE MODELS

**Metrics** We employ two metrics to examine the math capabilities of language models.

The first is a Pass@N metric

$$\text{Pass@N} = \mathop{\mathbb{E}}_{\text{Problems}} \left[ \min(c, 1) \right], \tag{1}$$

where $c$ represents the number of correct answers out of $N$ responses. This metric considers a question to be solved if at least one correct answer is produced from $N$ random generations. We employ this metric to reflect the potential or capability of a model in solving a math question. To enhance the diversity of the $N$ generations, we set the temperature of the generation process to 0.7.

The second is a PassRatio@N metric

$$\text{PassRatio@N} = \mathop{\mathbb{E}}_{\text{Problems}} \left[ \frac{c}{N} \right], \tag{2}$$

which measures the percentage of correct answers within the $N$ generated answers.

When assessing the correctness of responses, the usual practice is to employ automated evaluation scripts to simply verify whether the model's final answer matches the ground truth. However, since it can only evaluate the final answer, misjudgments are inevitable. Given that GPT-4 Turbo's result on the GSM8K benchmark is very close to perfect, we believe it can provide a more accurate measurement of model performance. Therefore, we have augmented the LLaMA-2-7B's GSM8K Pass@N evaluation by an additional assessment of the intermediate reasoning using GPT-4 Turbo. Meanwhile, due to GPT4 Turbo's poorer performance and numerous observed misjudgments on the MATH benchmark during our trial, we don't adopt this evaluation method on MATH.

**Observations** Based on these two metrics, we examine the performance of the LLaMA-2 models on the GSM8K and the MATH benchmarks[2] as shown in Figure 1.

We first observe that the Pass@1024 metrics with only ground truth evaluation for the LLaMA-2 7B model on both benchmarks are remarkably high: 97.6% on GSM8K and 70.0% on MATH. When using GPT-4 Turbo as an additional judge, the Pass@256 on GSM8K goes to 91.8%. Moreover, as the

---

[1]Concurrently, DeepSeek-MATH-7B Shao et al. (2024) also surpasses 80% accuracy. However, their approach relies on a much stronger base model extensively pre-trained on math-related corpora and a sophisticated RL algorithm. Our results are complementary to theirs.

[2]Following Lightman et al. (2023), we utilize a subset of 500 test samples from the MATH benchmark when calculating Pass@N and PassRatio@N metrics for experimental efficiency.

number of N increases, the Pass@N results continue to show significant improvement. This suggests that the LLaMA-2 7B model possesses a strong capability for solving mathematical problems.

We then notice that the PassRatio@1024 is significantly lower than that of Pass@1024, being 9.3% on GSM8K and 2.8% on MATH. This suggests that while the correct answers to most math questions are present within 1024 generations, there is no assurance that the correct answers will consistently be extracted, a phenomenon we refer to as an *ranking issue*. In the following, we will present a simple approach to significantly reduce this issue.

# 3 SCALING SFT DATA USING SYNTHETIC MATH QUESTIONS

In this section, we first demonstrate that scaling up the limited real SFT data can significantly alleviate the ranking issue. We also observe that the accuracy has not yet plateaued when using the full available GSM8K and MATH training data. We consider further scaling up SFT data using synthetic math questions. To this aim, we introduce a straight-forward method for synthetic data generation utilizing the GPT-4 Turbo API. The synthetic data proves to be as effective as real math questions. Consequently, we boldly scale the synthetic SFT data to 960K on GSM8K and 480K on MATH, respectively, resulting in excellent scaling behavior, and reach state-of-the-art accuracy.

**Scaling using Real Math Questions** We begin by examining the scaling behavior of real math questions across the entire GSM8K and MATH training sets. As indicated in Table 1, we observe a consistent accuracy improvement, increasing from 26.7% to 50.2% on GSM8K, and from 4.2% to 8.4% on MATH, with no signs of saturation.

**Synthetic SFT Data Generation** Since the real data has been exhausted, we contemplate further scaling up SFT data using synthetic math questions. We introduce a straightforward three-step approach with the assistance of the GPT-4 Turbo API. The prompts are shown in Appendix A.

- *Step 1. Generate a new math question.* We request the GPT-4 Turbo API to generate a brand-new question using a reference math question as a starting point. To improve the validity of the new questions, we incorporate three rules into the prompt: Firstly, the new question must obey common knowledge; secondly, it should be solvable independently of the original question; and thirdly, it must not include any answer responses.

- *Step 2. Verify the question.* We further enhance the quality of the generated questions by validating and refining them through attempted solutions. By integrating solving and verification steps into a single prompt, we have found that this approach consistently elevates the validity of questions.

- *Step 3. Generate chain-of-thought (CoT) answers.* We request GPT-4 Turbo to produce a chain-of-thought (CoT) answer response for each newly generated question. Besides, we have set specific formatting requirements for answers tailored to various target datasets.

**Comparison of Synthetic SFT Data versus Real Data** To assess the quality of the synthetically generated math questions, we evaluate their effectiveness against real questions from the GSM8K and MATH training sets, utilizing a LLaMA-2 7B model, as detailed in Table 1. The results indicate that the synthetic math questions are nearly as effective as the real ones. We also explore various other synthetic methods as proposed in previous works Xu et al. (2023); Yu et al. (2023); An et al. (2023). These methods also prove to be effective, though marginally less so than our approach, as illustrated in Figure 6.

**Scaling to about a Million SFT Math Data** Considering the effectiveness of the synthetic approach, we substantially increase the scale of the SFT data for both GSM8K and MATH problems, to 960K and 480K, respectively. Figure 1 presents the main results utilizing various sizes of the LLaMA-2 series. The straightforward scaling strategy yields state-of-the-art accuracy. It is also worth noting that the accuracy has not yet reached its peak.

**Data Contamination Test** We evaluate benchmark leakage for the GSM8K dataset using 10-gram feature cosine similarity. Three questions from the test set have a similarity score above 0.35 with both the original and synthetic training sets. The maximum similarity in each of the two scenarios

is 0.81. For the MATH test set, 498 and 502 questions surpass a 0.5 similarity threshold when comparing with training set and synthetic set, respectively. This suggests our synthetic dataset does not significantly introduce additional leakage for GSM8K and MATH benchmarks.

## 4 EXPERIMENTS

### 4.1 IMPLEMENTATION DETAILS

In data synthesis, we utilize the GPT-4 Turbo API, setting the temperature to 1.0 for both question and answer generation. For supervised fine-tuning, we employ the Adam optimizer with a cosine learning rate schedule spanning a total of 3 epochs of training. The maximum learning rate is set to 2e-5 (except that 2e-6 for the Mistral-7b model) and there is a 4% linear warm-up. The maximum token length is set 2048, and the Vicuna-v1.1 Zheng et al. (2023) system prompt is used. All experiments are conducted on $8\times$Nvidia H100 GPUs. Our biggest experiment, involving a 70B model and 1440K data, takes 2800 H100 GPU hours. For generation, we use the same system prompt as used in SFT and set the maximum sequence length to 2048. The vLLM Kwon et al. (2023) is used in answer generation.

We conduct experiments on GSM8K Cobbe et al. (2021), MATH Hendrycks et al. (2021), SVAMP Patel et al. (2021), ASDiv Miao et al. (2021) and Hungarian National High School Math Exam xAI (2023) to evaluate the efficacy of the proposed method. It is worth noting that the score of Hungarian National High School Math Exam is judged by human, while other benchmarks are evaluated using automatic scripts, similar to previous works Luo et al. (2023); Gou et al. (2023).

### 4.2 MAIN RESULTS AND COMPARISON WITH STATE-OF-THE-ART MODELS

In this comparison, we examine both in-domain benchmarks, GSM8K/MATH, and out-of-domain benchmarks, such as the Hungarian National High School Math Exam. For in-domain evaluation of each benchmark, we utilize data synthesized from its respective training samples. For GSM8K, 960K synthetic data is employed, while for MATH, 480K synthetic data is used. For out-domain evaluation, we test models trained using GSM8K, MATH, or a mixed of two synthetic sets.

For base models, we consider both common language models, i.e., LLaMA-2 7B/13B/70B/Mistral-7B, and math-specific models, such as Llemma-7B, to assess the generality of the proposed approach.

Table 2: Math reasoning performances of various LLMs.

| Model | GSM8K | MATH |
|---|---|---|
| *Closed-source models* | | |
| GPT-4 Turbo (1106) | 94.8 | 64.5 |
| GPT-4-0314 | 94.7 | 52.6 |
| GPT-4 Achiam et al. (2023) | 92.0 | 42.5 |
| GPT-3.5-Turbo OpenAI (2023) | 80.8 | 34.1 |
| *Open-source models LLaMA-2-7B* | | |
| WizardMath-7B Luo et al. (2023) | 54.9 | 10.7 |
| MuggleMath-7B Li et al. (2023) | 68.4 | - |
| MetaMath-7B Yu et al. (2023) | 66.5 | 19.8 |
| LEMA-LLaMA-2-7B An et al. (2023) | 54.1 | 9.4 |
| Xwin-Math-7B (ours) | **82.4** | **40.1** |
| *Open-source models LLaMA-3-8B* | | |
| DART-Math-Llama3-8B Tong et al. (2024) | 81.1 | 46.6 |
| Xwin-Math-Llama3-8B (ours) | **87.9** | **52.3** |
| *Open-source models Mistral-7B* | | |
| WizardMath-7B-v1.1 Luo et al. (2023) | 83.2 | 33.0 |
| MetaMath-Mistral-7B Yu et al. (2023) | 77.4 | 28.2 |
| DART-Math-Mistral-7B Tong et al. (2024) | 81.1 | 45.5 |
| Xwin-Math-Mistral-7B (ours) | **88.2** | **48.4** |

| Model | GSM8K | MATH |
|---|---|---|
| *Open-source models Llemma-7B* | | |
| MetaMath-Llemma-7B Yu et al. (2023) | 69.2 | 30.0 |
| Xwin-Math-Llemma-7B (ours) | 83.1 | 45.5 |
| *Open-source models DeepSeekMath-7B* | | |
| DeepSeekMath-Instruct Yu et al. (2023) | 83.7 | **57.4** |
| DART-Math-DSMath-7B Tong et al. (2024) | 83.8 | 53.6 |
| Xwin-Math-Deepseekmath-7B (ours) | **88.9** | 55.4 |
| *Open-source models LLaMA-2-13B* | | |
| WizardMath-13B Luo et al. (2023) | 63.9 | 14.0 |
| MuggleMath-13B Li et al. (2023) | 74.0 | - |
| MetaMath-13B Yu et al. (2023) | 72.3 | 22.4 |
| LEMA-LLaMA-2-13B An et al. (2023) | 65.7 | 12.6 |
| Xwin-Math-13B (ours) | **87.6** | **44.4** |
| *Open-source models LLaMA-2-70B* | | |
| WizardMath-70B Luo et al. (2023) | 81.6 | 22.7 |
| MuggleMath-70B Li et al. (2023) | 82.3 | - |
| MetaMath-70B Yu et al. (2023) | 82.3 | 26.6 |
| LEMA-LLaMA-2-70B An et al. (2023) | 83.5 | 25.0 |
| Xwin-Math-70B (ours) | **90.6** | **51.9** |

**In-Domain Results** Table 2 presents a comparison of the proposed approach with the state-of-the-art open and closed-source models. The Xwin-Math models here are all trained with the mix of GSM8K and MATH synthetic data, leading to slightly different results from scaling curve in Figure 1. Across all base models, our method significantly outperforms the previous best approaches.

On LLaMA-2-7B, our approach exceeds the prior best by absolutely +14.0 on GSM8K (compared to MuggleMath-7B Li et al. (2023)), and by +20.3 on MATH (compared to MetaMath-7B Yu et al. (2023)), respectively. It even surpasses GPT-3.5 Turbo and several latest 70B models dedicated for math capabilities, such as WizardMath-70B (Luo et al., 2023) (40.1 versus 22.7 on MATH). On LLaMA-2-70B, the gains are +7.1 on GSM8K (compared to LEMA-LLaMA-2-70B An et al. (2023)) and +25.3 on MATH (compared to MetaMath-70B Yu et al. (2023)), respectively. On a stronger common language model, i.e., Mistral-7B, the improvements are +5.0 on GSM8K and +15.4 on MATH (compared to WizardMath-7B-v1.1 Luo et al. (2023)), respectively. On a math-specific base model, such as Llemma-7B, the gains are +13.9 on GSM8K and +15.5 on MATH (compared to MetaMath-Llemma-7B Luo et al. (2023)). The results based on LLaMA-3 and DeepSeekMath also shows our work is also applicable to these newly enhanced base models.

It is also noteworthy that our LLaMA-2-70B model achieves competitive accuracy with early versions of GPT-4 on GSM8K and MATH. To our knowledge, this is the first LLaMA2-based model to approach GPT-4-0314 on MATH. These results demonstrate the significant effectiveness and broad applicability of scaling synthetic math SFT data.

Figure 2: Hungarian National High School Math Exam test result of various LLMs.

| Model | Test Score (%) |
|---|---|
| GPT-4 Achiam et al. (2023) | 68 |
| Grok-1 xAI (2023) | 59 |
| DeepSeek-LLM-67B-Chat Bi et al. (2024) | 58 |
| Claude-2 Anthropic (2023) | 55 |
| GPT-3.5 Turbo OpenAI (2023) | 41 |
| Xwin-Math-70B (480K GSM8K) | 22 |
| Xwin-Math-70B (120K MATH) | 51 |
| Xwin-Math-70B (480K MATH) | 59 |
| Xwin-Math-70B (240K GSM8K+240K MATH) | 65 |
| Xwin-Math-70B (960K GSM8K+480K MATH) | 75 |

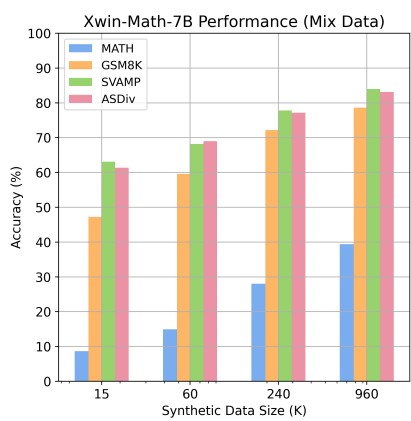

Figure 3: The performance on in-domain and out-of-domain metrics.

**Out-of-Domain Results** We test the models trained using GSM8K, MATH, or a mixed of two synthetic sets on an out-of-domain benchmark, Hungarian National High School Math Exam, following the practice in xAI (2023).

Table 2 shows the results. Our model trained on the mixing data (960K MATH synthetic data + 480K GSM8K synthetic data) preforms much better than other models, even an early version of GPT-4. Additionally, we plot the correlation between GSM8K and Hungarian National High School Math Exam scores in Appendix B. Also note that as the synthetic data scales up, the performance of the model on this out-of-domain benchmark continually increases. The results show that our method can enhance the general math capabilities rather than simply overfit the GSM8K and MATH benchmark.

Figure. 3 presents the results of the model trained on synthetic data using a mixture of GSM8K and MATH in a 1:1 ratio. We find that the accuracy of other out-of-domain benchmarks (SVAMP and ASDiv) also improves as the amount of data increases for models trained with synthetic data. These models exhibit balanced scaling behaviors in both in-domain and out-of-domain benchmarks.

Table 3: The performance of Xwin-Math-7B on more out-of-domain benchmarks.

| Model Name | Base Model | GSM-hard(%) | GSM-plus(%) |
|---|---|---|---|
| Xwin-Math-7B | LLaMA-2 7B | 41.2 | 61.1 |
| MetaMath-7B | LLaMA-2 7B | 20.8 | 44.2 |
| MetaMath-70B | LLaMA-2 70B | 45.5 | 60.4 |
| GPT-3.5-Turbo | - | - | 61.2 |

We conduct further experiments on two out-of-domain metrics related to GSM8K. The results on GSM-hard and GSM-plus in Tablw 3 show that our 7B model can still perform at the same level as the previous 70B open source model and GPT-3.5-Turbo.

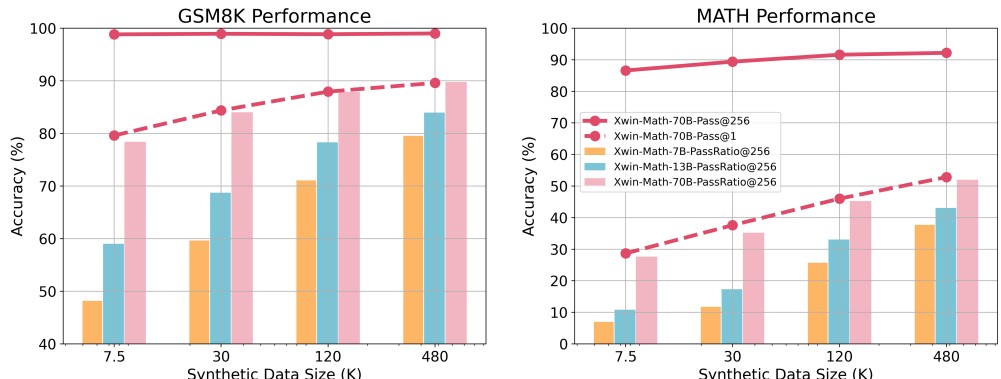

Figure 4: The Pass@256 and PassRatio@256 performance with increasing data scale on GSM8K and MATH benchmark.

### 4.3 WHAT HAPPENS BEHIND PERFORMANCE IMPROVEMENTS?

**Pass@256 *v.s.* PassRatio@256** To deepen the understanding behind the performance improvements, we track Pass@N metric and PassRatio@N metric under different data size. The results are shown in Figure 4. With very limited synthetic data (*e.g.* 7.5K samples), the Xwin-Math-70B model already has very high Pass@256, indicating the strong ability to generate correct answers through multiple attempts. Meanwhile, the Pass@256 metric only change slightly with increasing the amount of used data. In contrast, PassRatio@256, which reflects the stability to generate correct answer, increases significantly with the amount of synthetic data, and its growth trend is similar to that of Pass@1. This result confirms our hypothesize that the performance improvements is mainly caused by the better stability in answer generation rather than stronger ability to answer the question.

**Estimated Stepwise Reasoning Accuracy** Because of the Chain-of-Thought (CoT) are adopted in inference, the process of answer mathematical problems is completed by a multi-step reasoning process. Therefore, we hypothesize that the increase in final answer accuracy can be interpreted by the improvement in stepwise reasoning accuracy. Based on this assumption, if one question can be theoretically answered by $s$ reasoning steps in CoT, the final answer accuracy can be approximate by the power function of the stepwise reasoning accuracy:

$$\text{Acc}_{\text{final}} = \text{Acc}_{\text{step}}^{s} \tag{3}$$

With this equation, step accuracy can be estimated from the final answer accuracy. We experiment on GSM8K. For each question in the test set, we use Xwin-Math 7B to generate 256 responses and use the number of steps in the CoT annotations as the theoretical CoT steps. We draw the mean accuracy and the fitted curve based on Equation. 3 in Figure 5 with different amount of synthetic data. The solid line is fitted using all seven points and Table 4 shows the estimated stepwise accuracy when using different amounts of data using all data points, and it can be seen that the stepwise accuracy improve significantly with more data.

However, when we fit based on Equation. 3 with the first four points, as shown in dashed lines, we find that the latter three points are significantly below the curve. We believe this phenomenon may be related to the smaller proportion of more complex problems in the training data. Therefore, we resample the 960K synthetic data according to the number of sentences in CoT solution. As can be seen from Figure 5 (right), when the proportion of complex problems is increased, the accuracy for simpler problems remains virtually unchanged, but the accuracy for more complex problems can be significantly improved. Moreover, the utilization of data resampling can increase the model's PassRatio@256 from 71.1 to 72.8. This experimental result provides new insights into data selection for mathematical reasoning tasks.

In addition, we further use the GPT-4 Turbo to find the position where the first step in our answer is wrong and normalized that position by the total number of steps in each answer. As the estimated stepwise accuracy gets higher, the normalized first error position is postponed.

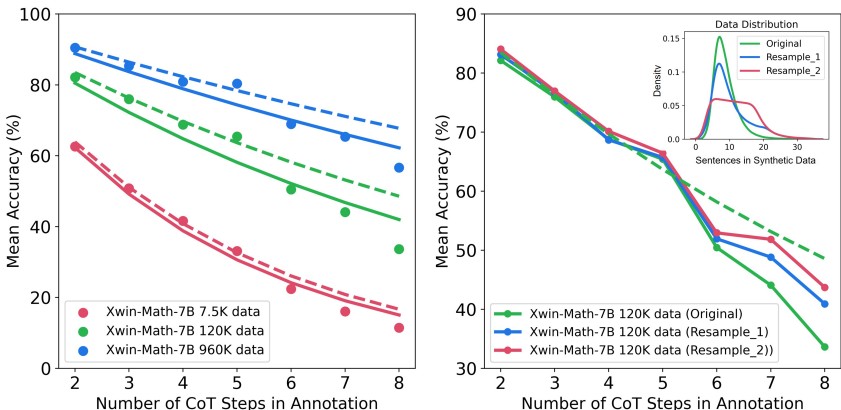

Figure 5: Left: The relationship between the mean accuracy on the GSM8K and the number of annotated CoT steps with data increasing. The solid line is fitted using all seven points, while the dashed line is fitted using the first four points. Right: Changes in mean accuracy when resampling is used to increase the CoT length of training data.

Table 4: The estimated stepwise reasoning accuracy and the average normalized first error position by GPT-4 Turbo in Xwin-Math-7B on GSM8K benchmark.

| Data size | Estimated Acc$_{step}$ (%) | Normalized first error position (%) |
|---|---|---|
| 7.5K | 78.9 | 67.1 |
| 120K | 89.7 | 83.9 |
| 960K | 94.2 | 90.9 |

### 4.4 ABLATIONS ON THE DATA SYNTHETIC SCHEMA

**Comparison with Other Data Synthetic Methods** We compare our approach with following common used data synthetic methods in WizardMath Luo et al. (2023), MuggleMath Li et al. (2023) and MetaMath Yu et al. (2023):

1) *Add Constraint.* Adding one more constrain while keeping others unchanged. 2) *Change Numbers.* Changing the numbers while keeping the context intact. 3) *Change Background.* Changing the background while keeping others the same. 4) *The Combination of Changing Numbers and Background.* A hybrid approach that combines changing both numbers and background. 5) *MetaMath Approach.* The synthetic methods proposed in MetaMath Yu et al. (2023), including answer augmentation, rephrasing question, self-verification question and FOBAR question. We follow their implementation but use GPT-4 Turbo to generate response data using their released questions.

The experimental results in the Figure 6 show that when the data size is relatively small, *e.g.*, 7.5k and 30k samples, the performance gap between the different methods is negligible. However, as the data size increases, our method and the method with added constraints show stronger performance. This suggests that the choice of data synthetic strategy becomes more critical as the data size increases, and that our method can scale the data more efficiently, thus improving the performance.

**The Diversity of Synthetic Data.** In order to measure the diversity of our synthetic data, we refer to the diversity gain metric used by MetaMath Yu et al. (2023). The results in Table 7 show that when the scale of synthetic data continues to increase, our data always maintains a relatively high diversity. This provides an explanation for the scale effect in our work.

Table 7: The diversity gain of Xwin-Math's synthetic GSM8K data compared with MetaMath.

| GSM8K-syn Data Size | 20K | 40K | 60K | 80K | 100K |
|---|---|---|---|---|---|
| Xwin-Math | 0.30 | 0.27 | 0.25 | 0.24 | 0.23 |
| MetaMath | 0.17 | 0.12 | 0.10 | 0.10 | 0.10 |

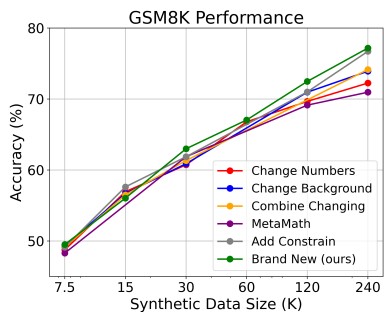

Figure 6: GSM8K performance of different synthetic methods.

Table 5: Ablation of question verification on MATH.

| Model | Pass@1 (%) |
|---|---|
| Xwin-Math-70B (7.5K data) | 28.9 |
| Xwin-Math-70B (7.5K data) w/o verification | 28.1 (-0.8) |
| Xwin-Math-70B (30K data) | 37.6 |
| Xwin-Math-70B (30K data) w/o verification | 36.6 (-1.0) |

Table 6: Benchmark leakage test

| Dataset | $L_{\text{test-regen}}$ | $L_{\text{test-ref}}$ | $L_{\text{train-regen}}$ | $\Delta_1$ | $\Delta_2$ |
|---|---|---|---|---|---|
| GSM8K | 0.52 | 0.50 | 0.33 | 0.02 | 0.19 |
| MATH | 0.59 | 0.58 | 0.39 | 0.01 | 0.20 |

**Effects of Question Verification.** The question verification is used to further improve the generation quality. In our experiments, we found it can improve the performance on MATH benchmark, the results are shown in Table 5, while we do not see significantly impact on GSM8K dataset.

### 4.5 BENCHMARK LEAKAGE TEST

To validate benchmark leakage during data generation, we compare LM loss on: 1) a regenerated synthetic training subset, where we maintain the original questions from the synthetic training subset and use GPT-4 Turbo to rewrite answers; 2) a regenerated test set, where we keep the test questions unchanged and use GPT-4 Turbo to rewrite answers; 3) a reference test set, where we use the test set as seed to generate new questions and answers via GPT-4 Turbo. Referring to Skywork Wei et al. (2023), we also report two key metrics: $\Delta_1 = L_{\text{test-regen}} - L_{\text{test-ref}}, \Delta_2 = L_{\text{test-regen}} - L_{\text{train-regen}}$, As $\Delta_1$ is close to 0 and $\Delta_2$ is significantly greater than 0 in two benchmarks, we believe that there is no significant leakage during the process of data synthesis.

## 5 RELATED WORKS

**Large Language Models** Large language models Brown et al. (2020); Achiam et al. (2023); Touvron et al. (2023a;b) have made significant achievements, with impressive performance on a wide range of tasks. Currently, closed-source large language models, represented by GPT Brown et al. (2020); Achiam et al. (2023), Gemini Team et al. (2023), Grok xAI (2023), and Claude-2 Anthropic (2023), are the most advanced models in terms of performance. However, open-source models, represented by LLaMA Touvron et al. (2023a), LLaMA-2 Touvron et al. (2023b) and Mixtral Jiang et al. (2024), have also progressed rapidly, and have even shown competitive performance with the closed-source models on some tasks. Our work, which aims to improve the performance of open-source LLMs on mathematical tasks by fine-tuning them on synthetic data.

**Reasoning Framework for Improving Mathematical Capability** Chain-of-thoughts Wei et al. (2022b) encourages the LLMs perform multi-step reasoning by specific designed prompts and can improve reasoning performance. Based on this work, many subsequent works suggesting further improvements Fu et al. (2022); Zhang et al. (2022); Kojima et al. (2022). The above works focus primarily on how to improve performance through better prompt design or inference strategies without fine-tuning the model, whereas our work focuses on how to improve the model itself, and thus these approaches are complementary to ours.

**Fine-tuned LLM for Math** Another sort of works Lightman et al. (2023); Luo et al. (2023); Azerbayev et al. (2023); Yue et al. (2023); Yu et al. (2023); An et al. (2023); Li et al. (2023); Gou et al. (2023) try to improve performance directly by training the model on mathematical data. A direct way is to use fine-tuning to improve models. One widely used method is to use synthetic data, which is very close to our approach: MetaMath Yu et al. (2023) presents to bootstrap questions to augment data. LeMA An et al. (2023) collects mistake-correction data pairs by using GPT-4 as a corrector. And MuggleMath Li et al. (2023) augments the GSM8K dataset by incorporating GPT-4 with a series of pre-defined operations. Compared to these synthetic data based efforts, our data synthetic method is much simpler and more scalable.

**SFT Data Scaling** Recently, some research efforts have focused on the data scale for supervised fine-tuning. For instance, LIMA Zhou et al. (2023) mentions that fine-tuning with 1,000 high-quality instructions can yield impressive results in various general tasks. Other studies have indicated that performance scales with data size in mathematical and coding tasks Dong et al. (2023). Recent work Bi et al. (2024) even uses 1.5 million data for instruct fine-tuning to obtain top performance. However, the intrinsic reasons behind this scaling effect have not been thoroughly investigated.

## 6 CONCLUSION

This study reveals that common 7B language models, such as LLaMA-2 7B, already exhibit strong mathematical capabilities, challenging the previous belief. By significantly scaling up SFT data, we have markedly improved the stability of the model's mathematical problem-solving skills. Our methodology has enabled the Xwin-Math models to reach performance levels comparable to, and in some instances surpassing, those of their larger counterparts. In out-of-domain evaluation, our model also surpassed the performance of GPT-4, indicating that our methodology enhances the general mathematical reasoning capabilities. Our analysis also indicates that the enhancements are primarily attributable to heightened accuracy in reasoning and a extra resampling of training data can improve the accuracy of harder questions. Our research contributes valuable insights into the mathematical capabilities of large language models.

## 7 LIMITATIONS AND SOCIAL IMPACT

While our work demonstrates impressive performance in mathematics reasoning, generating synthetic data incurs significant computational expenses. Although the proposed method may have the potential to generalize to other capabilities, the computational costs increase linearly as the amount of data scales up. Meanwhile, since our method fundamentally relies on supervised fine-tuning, the associated training costs become significant when scaling up. Using OpenAI's API to synthesize data with GPT-4 Turbo costs approximately $100,000. Although this expense could decrease as API service continues to become cheaper, this remains a significant expenditure for researchers. On the other hand, our data scaling method has the potential to introduce noise during the generation of new data, which could constrain the effectiveness of the method depending on the model employed for data generation. Furthermore, the utilization of pre-trained language models in this work may entail certain ethical risks similar to those associated with other deep learning models, such as biases and privacy violations.

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

# A SYNTHETIC PROMPTS

## A.1 GSM8K

---
**Prompt 1: Question Generation**

Please act as a professional math teacher.

Your goal is to create high quality math word problems to help students learn math.

You will be given a math question. Please create a new question based on the Given Question and following instructions.

To achieve the goal, you have three jobs.

# Please generate a similar but new question according to the Given Question.

# Check the question by solving it step-by-step to find out if it adheres to all principles.

# Modify the created question according to your checking comment to ensure it is of high quality.

You have five principles to do this.

# Ensure the new question only asks for one thing, be reasonable, be based on the Given Question, and can be answered with only a number (float or integer). For example, DO NOT ask, 'what is the amount of A, B and C?'.

# Ensure the new question is in line with common sense of life. For example, the amount someone has or pays must be a positive number, and the number of people must be an integer.

# Ensure your student can answer the new question without the given question. If you want to use some numbers, conditions or background in the given question, please restate them to ensure no information is omitted in your new question.

# Please DO NOT include solution in your question.

# If the created question already follows these principles upon your verification. Just keep it without any modification.

Given Question: given question

Your output should be in the following format:

CREATED QUESTION: <your created question>

VERIFICATION AND MODIFICATION: <solve the question step-by-step and modify it to follow all principles>

FINAL CREATED QUESTION: <your final created question>

---

---
**Prompt 2: Answer Generation**

Please act as a professional math teacher.
Your goal is to accurately solve a math word problem.
To achieve the goal, you have two jobs.
# Write detailed solution to a Given Question.
# Write the final answer to this question.
You have two principles to do this.
# Ensure the solution is step-by-step.
# Ensure the final answer is just a number (float or integer).
Given Question: given question
Your output should be in the following format:
SOLUTION: <your detailed solution to the given question>
FINAL ANSWER: <your final answer to the question with only an integer or float number>

---

**Prompt 3: Question Generation w/o verification**

Please act as a professional math teacher.

Your goal is to create high quality math word problems to help students learn math.

You will be given a math question. Please create a new question based on the Given Question and following instructions.

To achieve the goal, you have one job.

# Please generate a similar but new question according to the Given Question.

You have four principles to do this.

# Ensure the new question only asks for one thing, be reasonable, be based on the Given Question, and can be answered with only a number(float or integer). For example, DO NOT ask, 'what is the amount of A, B and C?'.

# Ensure the new question is in line with common sense of life. For example, the amount someone has or pays must be a positive number, and the number of people must be an integer.

# Ensure your student can answer the new question without the given question. If you want to use some numbers, conditions or background in the given question, please restate them to ensure no information is omitted in your new question.

# You only need to create the new question. Please DO NOT solve it.

Given Question: given question

Your output should be in the following format:

CREATED QUESTION: <your created question>

## A.2   MATH

**Prompt 1: Question Generation**

Please act as a professional math teacher.

Your goal is to create high quality math word problems to help students learn math.

You will be given a math question. Please create a new question based on the Given Question and following instructions.

To achieve the goal, you have three jobs.

# Please generate a similar but new question according to the Given Question.

# Check the question by solving it step-by-step to find out if it adheres to all principles.

# Modify the created question according to your checking comment to ensure it is of high quality.

You have five principles to do this.

# Ensure the new question only asks for one thing, be reasonable, be based on the Given Question, and can be answered with only a number (float or integer) or a LaTeX mathematical expression. For example, DO NOT ask, 'what is the amount of A, B and C?'.

# Ensure the new question is in line with common sense of life. For example, the amount someone has or pays must be a positive number, and the number of people must be an integer.

# Ensure your student can answer the new question without the given question. If you want to use some numbers, conditions or background in the given question, please restate them to ensure no information is omitted in your new question.

# Please DO NOT include solution in your question.

# If the created question already follows these principles upon your verification. Just keep it without any modification.

Given Question: given question

Your output should be in the following format:

CREATED QUESTION: <your created question>

VERIFICATION AND MODIFICATION: <solve the question step-by-step and modify it to follow all principles>

FINAL CREATED QUESTION: <your final created question>

## Prompt 2: Answer Generation

Please act as a professional math teacher.

Your goal is to accurately solve a math problem.

To achieve the goal, you have two jobs.

# Write detailed solution to a Given Question.

# Write the final answer to this question.

You have two principles to do this.

# Ensure the solution is step-by-step.

# Ensure the final answer is just a number (float or integer) or a LaTeX mathematical expression.

Given Question: given question

Your output should be in the following format:

SOLUTION: <your detailed solution to the given question>

FINAL ANSWER: <your final answer to the question with only a number (float or integer) or a LaTeX mathematical expression>

## Prompt 3: Question Generation w/o verification

Please act as a professional math teacher.

Your goal is to create high quality math word problems to help students learn math.

You will be given a math question. Please create a new question based on the Given Question and following instructions.

To achieve the goal, you have one job.

# Please generate a similar but new question according to the Given Question.

You have four principles to do this.

# Ensure the new question only asks for one thing, be reasonable, be based on the Given Question, and can be answered with only a number (float or integer) or a LaTeX mathematical expression. For example, DO NOT ask, 'what is the amount of A, B and C?'.

# Ensure the new question is in line with common sense of life. For example, the amount someone has or pays must be a positive number, and the number of people must be an integer.

# Ensure your student can answer the new question without the given question. If you want to use some numbers, conditions or background in the given question, please restate them to ensure no information is omitted in your new question.

# You only need to create the new question. Please DO NOT solve it.

Given Question: given question

Your output should be in the following format:

CREATED QUESTION: <your created question>

## B   ADDITIONAL RESULTS

Figure 7: Xwin-Math's aggregate performance on these two benchmarks is second only to GPT-4, demonstrating our model's robust generalization capabilities.

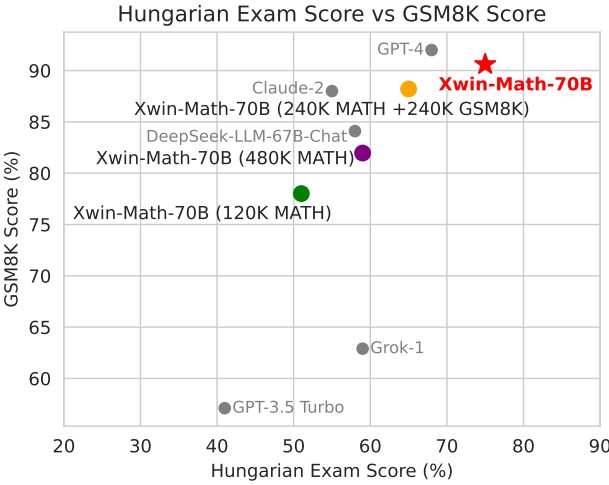

**The Improvement in the Accuracy of Numerical Calculations is More Significant than Logical Reasoning** The performance of the model gradually improves as the synthetic data increases. For a deeper understanding, we analyze the error proportion for different types of errors on GSM8K. We categorize errors into two types: reasoning errors and calculation errors. Reasoning errors primarily encompass issues such as loss of conditions and concept confusion, while calculation errors include incorrect analysis of quantitative relationships and numerical computation mistakes. Based on the experimental results illustrated in Figure 8, we observe a gradual decrease in the percentage of calculation errors, suggesting that GSM8K is correcting calculation errors at a faster rate than reasoning errors.

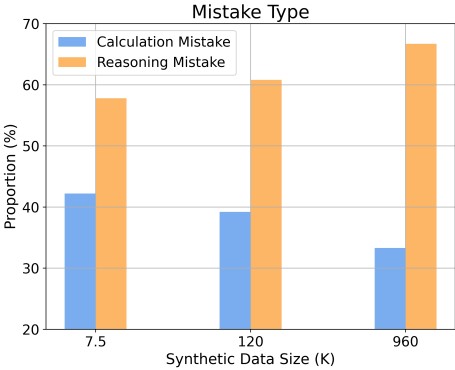

Figure 8: Changes in the proportion of calculation and reasoning mistake with increasing data scale.

## C  CASE STUDY OF SYNTHETIC DATA

### C.1  GSM8K

---

**Original Question**

Randy has 60 mango trees on his farm. He also has 5 less than half as many coconut trees as mango trees. How many trees does Randy have in all on his farm?

---

**Synthetic Question w/o Verification**

Maria has 80 apple trees in her orchard. She also has 4 less than a third as many cherry trees as apple trees. How many trees does Maria have in all in her orchard?

---

**Verification Comment**

Let's check the created question:

- Maria has 80 apple trees.

- She has 4 less than a third as many cherry trees as apple trees. A third of the apple trees is 80/3, which is approximately 26.67.

- Four less than that would be around 22.67, but since we cannot have a fraction of a tree, we round it to the nearest whole number, which is 27.

- The modified question should ask for the number of cherry trees without the decimal. So we adjust the question to have a whole number as a result.

- Since 80 is not divisible by 3, to get a whole number of cherry trees, we can change the number so that it is divisible by 3. Let's use 81 apple trees instead of 80.

---

**Synthetic Question with Verification**

Maria has 81 apple trees in her orchard. She also has 4 less than a third as many cherry trees as apple trees. How many trees does Maria have in all in her orchard?

---

**Synthetic Solution and Final Answer**

First, let's determine the number of cherry trees Maria has. We are told she has 4 less than a third as many cherry trees as apple trees.

Since Maria has 81 apple trees, let's find one third of that number:

1/3 of 81 apple trees = 81 apple trees / 3 = 27 apple trees.

Now, since she has 4 less than this amount in cherry trees, we subtract 4 from 27 to find the number of cherry trees:

27 cherry trees - 4 = 23 cherry trees.

To find the total number of trees Maria has in her orchard, we add the number of apple trees to the number of cherry trees:

81 apple trees + 23 cherry trees = 104 trees.

The answer is 104 .

## C.2 MATH

**Original Question**

The value of $\log_{10} 28471$ is between the consecutive integers $a$ and $b$. Find $a + b$.

**Synthetic Question w/o Verification**

The value of $\log_2 16384$ is between the consecutive integers $x$ and $y$. Find $x + y$.

**Verification Comment**

To ensure the question can be solved, we need to check if the logarithm given can be evaluated and falls between two consecutive integers.
1. Identify the value of $2^n$ that equals $16384$.
2. If $16384$ is a power of 2, then $\log_2 16384$ will be an integer.
3. Check to ensure the result is an integer and find the consecutive integers around it.
Let's perform the check:
$2^0 = 1, 2^1 = 2, 2^2 = 4, 2^3 = 8, 2^4 = 16$
...
$2^{14} = 16384$
So, $\log_2 16384 = 14$.
The consecutive integers around 14 are (13 and 14) or (14 and 15).
Since the logarithm gives an exact integer, to make the question valid, we will modify it to use a number that is not an exact power of 2.
MODIFIED CREATED QUESTION:
The value of $\log_2 10000$ is between the consecutive integers $m$ and $n$. Find $m + n$.
Now, let's check the modified question:
1. Since $10000$ is not a power of 2, $\log_2 10000$ will not be an integer.
2. The goal is to find two consecutive integers between which this logarithm value lies.
3. Knowing that $2^{13} = 8192$ and $2^{14} = 16384$, $\log_2 10000$ is between 13 and 14.
The consecutive integers around $\log_2 10000$ are 13 and 14.

**Synthetic Question with Verification**

The value of $\log_2 10000$ is between the consecutive integers $m$ and $n$. Find $m + n$.

**Synthetic Solution and Final Answer**

To find the consecutive integers between which $\log_2 10000$ lies, we need to find the powers of 2 that are immediately less than and greater than $10000$.

First, let's find the largest power of 2 that is less than $10000$. Starting with $2^1, 2^2, \ldots$ and so on, we find:

$2^{10} = 1024, 2^{11} = 2048, 2^{12} = 4096, 2^{13} = 8192, 2^{14} = 16384$

Here, we see that $2^{13} < 10000 < 2^{14}$, which means that $\log_2 10000$ is greater than 13 but less than 14.

Therefore, the consecutive integers $m$ and $n$ between which $\log_2 10000$ lies are $m = 13$ and $n = 14$.

Now we will add these two integers to find the required sum:

$m + n = 13 + 14 = 27$.

The answer is $\boxed{27}$.

# D CASE STUDY OF MODEL RESPONSE

## D.1 GSM8K

---

**Question and Ground Truth Solution**

**Question:** A treasure hunter found a buried treasure chest filled with gems. There were 175 diamonds, 35 fewer rubies than diamonds, and twice the number of emeralds than the rubies. How many of the gems were there in the chest?

**Solution:** There were 175 - 35 = 140 rubies in the treasure chest. There were 140 x 2 = 280 emeralds in the treasure chest. So, the total number of gems in the treasure chest is 175 + 140 + 280 = $\boxed{595}$.

---

**Response – Xwin-Math-7B (7.5k data) Greedy Decode**

First, let's find out the number of rubies. We are told there are 35 fewer rubies than diamonds. Since there are 175 diamonds, we can calculate the number of rubies as follows:
Number of rubies = Number of diamonds - 35 = 175 - 35 = 140
Next, we need to find out the number of emeralds. We are told there are twice the number of emeralds than the rubies. So we can calculate the number of emeralds as follows:
Number of emeralds = 2 × Number of rubies = 2 × 140 = 280
Now, we can find the total number of gems by adding the number of diamonds, rubies, and emeralds together:
Total number of gems = Number of diamonds + Number of rubies + Number of emeralds = 175 + 140 + 280 = 605
The answer is: $\boxed{605}$. **Incorrect** ⟶ **Wrong Calculation**

---

**Response – Xwin-Math-7B (7.5k data) One Sample from Temperature=0.7**

Let's find the number of each type of gem in the chest step by step.
First, we are given the number of diamonds:
Number of diamonds = 175
Next, we are told there are 35 fewer rubies than diamonds. To find the number of rubies, we subtract 35 from the number of diamonds:
Number of rubies = Number of diamonds - 35 = 175 - 35 = 140
Then, we are told there are twice the number of emeralds than the rubies. To find the number of emeralds, we multiply the number of rubies by 2: Number of emeralds = 2 * Number of rubies = 2 * 140 = 280
Now, we add the number of diamonds, rubies, and emeralds together to find the total number of gems in the chest:
Total number of gems = Number of diamonds + Number of rubies + Number of emeralds = 175 + 140 + 280 = 595
The answer is: $\boxed{595}$. **Correct**

---

**Response – Xwin-Math-7B (960k data) Greedy Decode**

Let's break down the information given in the problem step by step.
1. We know there are 175 diamonds.
2. There are 35 fewer rubies than diamonds. Since there are 175 diamonds, we can calculate the number of rubies by subtracting 35 from 175.
Number of rubies = Number of diamonds - 35 = 175 - 35 = 140
3. There are twice the number of emeralds than the rubies. We can find the number of emeralds by multiplying the number of rubies by 2.
Number of emeralds = 2 × Number of rubies = 2 × 140 = 280
Now, to find the total number of gems in the chest, we add the number of diamonds, rubies, and emeralds together.
Total number of gems = Number of diamonds + Number of rubies + Number of emeralds = 175 + 140 + 280 = 595
The answer is: $\boxed{595}$. **Correct**

---

## D.2 MATH

---

**Question and Ground Truth Solution**

**Question:** If $f(x) = \frac{3x-2}{x-2}$, what is the value of $f(-2) + f(-1) + f(0)$? Express your answer as a common fraction.

**Solution:** $f(-2) + f(-1) + f(0) = \frac{3(-2)-2}{-2-2} + \frac{3(-1)-2}{-1-2} + \frac{3(0)-2}{0-2} = \frac{-8}{-4} + \frac{-5}{-3} + \frac{-2}{-2} = 2 + \frac{5}{3} + 1 = \boxed{\frac{14}{3}}$

---

**Response – Xwin-Math-7B (7.5k data) Greedy Decode**

To find the value of $f(-2) + f(-1) + f(0)$, we need to evaluate the function $f(x)$ at these three values of $x$.

First, let's find $f(-2)$: $f(x) = \frac{3x-2}{x-2}$ Plug in $x = -2$: $f(-2) = \frac{3(-2)-2}{-2-2} = \frac{-6-2}{-4} = \frac{-8}{-4}$

Now, let's find $f(-1)$: Plug in $x = -1$: $f(-1) = \frac{3(-1)-2}{-1-2} = \frac{-3-2}{-3} = \frac{-5}{-3}$

Finally, let's find $f(0)$: Plug in $x = 0$: $f(0) = \frac{3(0)-2}{0-2} = \frac{-2}{-2} = 1$

Now, we can find the sum: $f(-2) + f(-1) + f(0) = \frac{-8}{-4} + \frac{-5}{-3} + 1$

To add these fractions, we need to find a common denominator, which is the least common multiple (LCM) of the three denominators (-4, -2, and -3). The LCM of these denominators is -12. **Unreasonable**

Now, we can rewrite each fraction with the common denominator:

$\frac{-8}{-4} = \frac{4}{3}$, $\frac{-5}{-3} = \frac{5}{3}$, $1 = \frac{12}{12}$

Now, we can add the fractions: $\frac{4}{3} + \frac{5}{3} + \frac{12}{12} = \frac{4+5+12}{3} = \frac{19}{3}$

The answer is: $\boxed{\frac{19}{3}}$. **Incorrect**   **Wrong calculation**

---

**Response – Xwin-Math-7B (7.5k data) One Sample from Temperature=0.7**

To find the value of $f(-2) + f(-1) + f(0)$, we first need to evaluate each function at these specific values of x.

For $f(-2)$: $f(x) = \frac{3x-2}{x-2}$ Substitute $x = -2$ into the equation: $f(-2) = \frac{3(-2)-2}{-2-2} = \frac{-6-2}{-4} = \frac{-8}{-4} = 2$

For $f(-1)$: $f(x) = \frac{3x-2}{x-2}$ Substitute $x = -1$ into the equation: $f(-1) = \frac{3(-1)-2}{-1-2} = \frac{-3-2}{-3} = \frac{-5}{-3} = \frac{5}{3}$

For $f(0)$: $f(x) = \frac{3x-2}{x-2}$ Substitute $x = 0$ into the equation: $f(0) = \frac{3(0)-2}{0-2} = \frac{-2}{-2} = 1$

Now we add the values of $f(-2)$, $f(-1)$, and $f(0)$: $f(-2) + f(-1) + f(0) = 2 + \frac{5}{3} + 1 = \frac{14}{3}$

The answer is: $\boxed{\frac{14}{3}}$. **Correct**

---

**Response – Xwin-Math-7B (480k data) Greedy Decode**

To find the value of $f(-2) + f(-1) + f(0)$, we need to evaluate the function $f(x)$ at each of the given points and then sum the results.

First, let's find $f(-2)$: $f(-2) = \frac{3(-2)-2}{(-2)-2} = \frac{-6-2}{-2-2} = \frac{-8}{-4} = 2$

Next, let's find $f(-1)$: $f(-1) = \frac{3(-1)-2}{(-1)-2} = \frac{-3-2}{-1-2} = \frac{-5}{-3} = \frac{5}{3}$

Finally, let's find $f(0)$: $f(0) = \frac{3(0)-2}{0-2} = \frac{0-2}{-2} = \frac{-2}{-2} = 1$

Now, we can sum these values: $f(-2) + f(-1) + f(0) = 2 + \frac{5}{3} + 1$ To add these, we need a common denominator, which is 3: $2 + \frac{5}{3} + 1 = \frac{2 \cdot 3}{3} + \frac{5}{3} + \frac{1 \cdot 3}{3} = \frac{6}{3} + \frac{5}{3} + \frac{3}{3}$

Now, we add the numerators: $\frac{6}{3} + \frac{5}{3} + \frac{3}{3} = \frac{6+5+3}{3} = \frac{14}{3}$

The answer is: $\boxed{\frac{14}{3}}$. **Correct**

