# OpenReview forum: "Common 7B Language Models Already Possess Strong Math Capabilities"
_ICLR.cc/2025/Conference — ICLR 2025 Conference Withdrawn Submission_

### Official Review · Reviewer_Y5EL · 2024-11-02

**Soundness:** 2
**Presentation:** 3
**Contribution:** 1
**Rating:** 3
**Confidence:** 4

**Summary:**

The paper investigates the mathematical capabilities of LLMs with a focus on GSM8K and MATH datasets. First, the authors show that with a large number of response generations (e.g., generating 1024 responses), the accuracy, measured by the probability of at least one generated response being correct (e.g., pass@1024), improves significantly. The paper claims that the presence of correct answers within these 1024 generated responses suggests that the model already possesses strong math abilities, and the observed performance issues are due to instability in extracting responses, which the paper refers to as a "ranking issue". To alleviate this, the authors propose using an external LLM (e.g., GPT-4) to generate synthetic data for finetuning, which can increase accuracy on various math benchmarks.

**Strengths:**

1- Understanding reasoning capabilities of models, including mathematical capabilities, is an important research direction.
2- The paper is well-written and the arguments are presented well.

**Weaknesses:**

1- The observation that increasing the number of generations improves the probability of obtaining a correct answer is not new to this work. For example, figure 4 from [1] reports the same observation on GSM8K.

2- Most importantly, the use of pass@1024 to claim a model's "capability" is methodologically questionable. Consider this extreme example for clarity: Imagine the target task is finding a path from start to end in a maze. If I write a program that randomly tests paths in the maze, with enough generations (depending on the maze size and difficulty), one of the outputs will be correct. Does that mean my program is capable of solving mazes? While LLMs don't output random tokens, and this is an extreme example for demonstration purposes, the overall situation is similar.

3- The paper assumes that higher accuracy on GSM8K indicates better 'mathematical reasoning' capabilities. Recent works such as [2] suggest that models do not understand mathematical concepts and rely on pattern matching and memorization, which could explain the reported improved results from using synthetic data for finetuning in this work. Note that the experiments in this paper does not provide any evidence for reasoning improvement, including the OOD experiments. For instance, the Hungarian National High School Math Exam is very similar to the MATH benchmark, and we observe from figure 2 that training on 120K synthetic examples from MATH leads to ~30% more accuracy than training on 480K synthetic examples from GSM8K. This raises questions about whether this benchmark is truly OOD for MATH, and whether the improvements reflect enhanced "mathematical reasoning" or simply exposure to similar questions in the synthetic data, especially given that we cannot rule out GPT-4's prior exposure to these questions.

4- Regarding the generation/use of synthetic data, there are no particularly new insights. There is no formal mechanism to guarantee the correctness of problems and solutions. In fact, Figure 8 of the paper (in Appendix) shows that the reasoning mistakes of the model increase as more synthetic data is used. This relates to my previous comments that there is no evidence (neither in the literature nor this work) that these LLMs possess "mathematical understanding/reasoning capabilities". However, there are several pieces of evidence, including this work, showing that the models rely on seeing similar questions and memorizing overall patterns to solve similar questions.

Overall, I find many of the paper's assumptions and claims to be incorrect, and the experiments do not adequately address these issues.

----
References:
[1] Shi, Chufan, et al. "A thorough examination of decoding methods in the era of llms." arXiv preprint arXiv:2402.06925 (2024).
[2] Mirzadeh, Iman, et al. "Gsm-symbolic: Understanding the limitations of mathematical reasoning in large language models." arXiv preprint arXiv:2410.05229 (2024).
[3] Zhang, Hugh, et al. "A careful examination of large language model performance on grade school arithmetic." arXiv preprint arXiv:2405.00332 (2024).

**Questions:**

1- For generating new questions, the authors use an LLM via prompts in the appendix, but it's unclear where the seed/original questions come from. For instance, for GSM8K, do the original questions come from the train set or test set?

---

### Official Review · Reviewer_pt3p · 2024-11-03

**Soundness:** 3
**Presentation:** 3
**Contribution:** 2
**Rating:** 3
**Confidence:** 5

**Summary:**

The authors present interesting findings that llama2-7b models can be fine-tuned to the level of SOTA closed source models on the MATH and GSM8K benchmarks when trained on large amounts of synthetic data from these SOTA models in a straight forward distillation process.

**Strengths:**

The paper has interesting empirical findings that may be of interest to researchers & developers on the math performance of small LLMs on large amounts of synthetic distillation data. The paper is generally well written and easy to follow, for the most part. The methods are detailed especially nicely to ensure ease of understanding and reproducibility.

**Weaknesses:**

The main weaknesses are:
1. Lack of significant novelty or surprising results: Prior work has shown that extensive math-related data pre-training improves the math capabilities of 7B scale models. These results, illustrated even w/in the first sentence of the abstract, reduces the novelty of this work. It follows that if 7B scale models have stronger math performance after pretraining on more math related data, that we can similarly distill strong model math perf into smaller models in post-training.
2. Lack of depth of analysis: you make an important observation that "The primary issue is the lack of guarantee that the correct answer will be dug out, as most generations are incorrect.", but this point is not deeply studied in the rest of the paper. The initial motivating observation that 7B models high have pass@1024 is under explored in empirical experiments, theoretical analysis, and final conclusion.
3. Severe lack of breadth:
 a. the lack of evaluation on llama3 is a glaring miss that the authors likely foresaw would arise during reviews. Given that the empirically strong performance of distilled llama2 is one of the strongest conclusions of this paper, running the analysis on llama3 is left as an obviously open question
b. The analysis conducted only on math and gsm8k. It's not clear how limited the author's claims are to these benchmarks. Further, section 4.3 would be much stronger using LLM or human stepwise grading or if an open source process-supervision dataset such as PRM800k could be leveraged.

**Questions:**

Below I've started w/ a few major questions and then follow w/ a number of small suggestions:
1. Will xwin and the associated dataset be open sourced? That would substantially increase the significance of this work and may influence my overall rating.
2. Where did the xwin name come from? what does it mean ?
3. lines 15-16: "achieved by selecting the oracle response from 1024 generations". You mean pass@k?Please try to use standard terminology or reword this part.
4. lines 16-17: "Equipped GPT-4 Turbo as an additional verification"  This has a grammatical error and doesn't make sense nor stand on it's own. Please correct the grammar on these lines, describe in a straightforward way, and improve the clarity of the methods section on this topic in the main body of the paper.
5. lines 48-51: you’re comparing pass@1024 to pass@1. Please clarify for readers.
6. line 100: “based on preference ones” doesn’t make sense
7. lines 106-107: "This data scaling shows excellent scaling behavior". The wording “excellent scaling behavior” is subjective and not defined. Please make clear what you mean here or move it to the discussion or remove it.
8. Eq 3: This equation is a nice observation and could be  connected more deeply via experiment or theory  to explain how training on synthetic drives the model to select the correct generation.
9. Section 4.3 and Figure 5: While an easy analysis, i could equally conclude that this is more about the model getting better at long problems than improving reasoning chains. Actual analysis of step wise accuracy via an LLM grader or humans or an existing dataset (like PRM800k if possible) would be  more conclusive

---

### Official Review · Reviewer_D8Z5 · 2024-11-04

**Soundness:** 1
**Presentation:** 1
**Contribution:** 2
**Rating:** 3
**Confidence:** 4

**Summary:**

This paper explores the mathematical capabilities of the LLaMA-2 7B model, suggesting that small language models can achieve high performance in mathematical reasoning when fine-tuned with synthetic data generated by GPT-4 Turbo. The LLaMA-2 7B model achieves 82.4% accuracy on the GSM8K benchmark and 40.1% on the MATH benchmark, outperforming some larger models, including earlier versions of GPT-4. By scaling up the supervised fine-tuning (SFT) data with GPT-4-generated questions, the authors claim that the synthetic data performs comparably to real data. The approach purportedly avoids performance saturation and shows competitive results across benchmarks.

**Strengths:**

Strengths:
Scaling Approach: The paper demonstrates that scaling synthetic data can effectively improve performance for smaller models, which challenges the assumption that only large-scale models excel in complex reasoning tasks.
High Benchmark Performance: The LLaMA-2 7B model's performance on GSM8K and MATH benchmarks is commendable and suggests that synthetic data can be a viable alternative to real data in specific contexts.
Experiment Setup: The paper includes various data scaling scenarios and compares performance against state-of-the-art models, adding insight into the practical applications of synthetic data for improving mathematical capabilities in language models.

**Weaknesses:**

1 Training Data Overlap Concerns: There’s a significant possibility that LLaMA-2's pretraining data includes or paraphrases the benchmarks used in evaluation, especially given the unknowns around its training corpus. This overlap could mean that the results simply "unlock" capabilities that were already in the model due to exposure during pretraining. Testing on a model where pretraining data is known and does not include the benchmarks would provide a more credible assessment.
2 Limited Benchmark Selection: The focus on GSM8K and MATH restricts evaluation to high school and grade school mathematics. More challenging, diverse benchmarks like MMLU (especially college-level sections) and OlympiadBench could provide a stronger test of the model's robustness. Success on OlympiadBench (IMO) [3], for example, would be a stronger indication of broader mathematical reasoning abilities.
3 Overstated Claims of Generality: The paper frequently implies that its findings apply to "common 7B LMs," yet the experiments focus solely on LLaMA-2 7B. Generalizing these results to other 7B models without further testing is premature.
4 No Discussion on Synthetic Data Risks: The paper lacks a discussion on "model collapse" from synthetic data, an issue documented in the literature [2]. While using GPT-4 may mitigate some risks, the absence of model degradation across such extensive data scaling is unusual. This aspect needs further exploration to explain the improvement consistency.
5 Large Data Volume Requirements: The use of 1024 generations per problem (yielding up to 76M examples) raises questions about the efficiency and feasibility of this approach. Such large data requirements, combined with the potential overlap between synthetic and benchmark data, make the setup less compelling.
6 Reliance on GPT-4 as Judge: The use of GPT-4 to evaluate generated responses without validating its correlation with human evaluations weakens the reliability of the reported improvements. A smaller-scale human evaluation or citation demonstrating alignment between GPT-4 judgments and human judgments would improve credibility.
7 Power Law Claim without Evidence: The claim of "power law scaling" is unsupported by a functional form or detailed analysis. Without a clearly defined functional relationship, the observed improvements should be framed as scaling trends, not power laws.
Reservations - Writing Concerns:
8 Misleading Introduction on Emergence: The introduction on emergence theory is confusing (It’s known emergence is caused by metrics now and in addition this is not how we grade people) and does not add value to the main narrative. It distracts from the paper's core contributions and should be revised or strongly suggested its removed.
9 Lack of Quantitative Detail in the Abstract: Statements like "significant improvement" are vague without absolute or relative performance changes. Quantitative results and implications should be included in the abstract to clearly communicate the impact of the findings.
10 Clarity and Readability: The paper would benefit from a thorough review by a native English speaker. There are multiple areas where phrasing is unclear or lacks precision, detracting from the scientific rigor of the presentation.
11 Captions and Figures: The captions do not provide sufficient context or summaries of the figures' main takeaways. Including summary values (like percentages) in the captions or directly in the figures would make it easier for readers to interpret the results.
12 “Table 2 shows the results.” This type of sentence raises major concerns about the writing of the whole paper.

**Questions:**

weaknesses

---

### Official Review · Reviewer_o81N · 2024-11-05

**Soundness:** 2
**Presentation:** 3
**Contribution:** 2
**Rating:** 5
**Confidence:** 3

**Summary:**

The paper asserted that  mathematical abilities of small-scale language models, LLaMA-2 7B, inherently possess strong mathematical capabilities. They prompt the pre-trained LLaMA-2 7B model to generate multiple answers (up to 1024) for each question in math benchmarks like GSM8K and MATH. By selecting the correct answer from these numerous attempts (an "oracle" approach), they find that the model can achieve high accuracy (97.6% on GSM8K and 70% on MATH). To improve consistency, they hypothesize that increasing the amount of supervised fine-tuning data can help the model learn to produce correct answers more reliably. They use a stronger language model, GPT-4 Turbo, to generate a large number of synthetic math problems and their solutions. However, the reference questions are from  GSM8K and  MATH. In other words, the new questions are similar to the questions in GSM8K and MATH. Which ends up with overfitting towards to solving GSM8K and MATH.

**Strengths:**

The authors uncover an interesting and less-explored phenomenon: by generating a large number of responses (up to 1024) per question, even small-scale language models like LLaMA-2 7B can achieve remarkably high accuracy on mathematical benchmarks.
Comprehensive Experimental Evaluation: The authors conduct extensive experiments across multiple benchmarks, including GSM8K, MATH, etc.
Well-Structured Presentation: The paper is organized logically, guiding the reader through the motivation, methodology—including the novel use of multiple response generations—results, and analysis in a coherent manner.

**Weaknesses:**

Lack of Novelty: The approach of enhancing language models using synthetic data and multiple answer generation has been explored in previous studies. And the way they generate the SFT data may cause overfitting towards the evaluation..
Practical Limitations: Generating 1024 responses per question to select the correct answer is impractical for real-world applications due to computational inefficiency.

**Questions:**

Potential Overfitting to Evaluation Benchmarks: Given the synthetic supervised fine-tuning (SFT) data generation process and its similarity to GSM8K and MATH benchmark questions, there is a concern that the model may be overfitting to these evaluation datasets. Could the authors elaborate on any measures taken to prevent overfitting, and whether they tested on additional, unrelated benchmarks to ensure that the model’s mathematical abilities generalize beyond GSM8K and MATH?

---

### Note · Authors · 2024-11-26

I have read and agree with the venue's withdrawal policy on behalf of myself and my co-authors.